# Preparation of Polylactic Acid/Calcium Peroxide Composite Filaments for Fused Deposition Modelling

**DOI:** 10.3390/polym15092229

**Published:** 2023-05-08

**Authors:** Abdullah H. Mohammed, Nikolina Kovacev, Amr Elshaer, Ammar A. Melaibari, Javed Iqbal, Hany Hassanin, Khamis Essa, Adnan Memić

**Affiliations:** 1School of Engineering, University of Birmingham, Birmingham B15 2TT, UK; 2Center of Nanotechnology, King Abdulaziz University, Jeddah 21589, Saudi Arabia; 3Drug Discovery, Delivery and Patient Care (DDDPC), School of Life Sciences, Pharmacy and Chemistry, Kingston University London, Kingston upon Thames KT1 2EE, UK; 4Department of Mechanical Engineering, King Abdulaziz University, Jeddah 21589, Saudi Arabia; 5School of Engineering, Canterbury Christ Church University, Canterbury CT1 1QU, UK

**Keywords:** FDM filaments, composite filaments, polylactic acid, calcium peroxide, 3D printing, biomedical filament

## Abstract

Fused Deposition Modelling (FDM) 3D printers have gained significant popularity in the pharmaceutical and biomedical industries. In this study, a new biomaterial filament was developed by preparing a polylactic acid (PLA)/calcium peroxide (CPO) composite using wet solution mixing and extrusion. The content of CPO varied from 3% to 24% wt., and hot-melt extruder parameters were optimised to fabricate 3D printable composite filaments. The filaments were characterised using an X-ray diffraction analysis, surface morphology assessment, evaluation of filament extrudability, microstructural analysis, and examination of their rheological and mechanical properties. Our findings indicate that increasing the CPO content resulted in increased viscosity at 200 °C, while the PLA/CPO samples showed microstructural changes from crystalline to amorphous. The mechanical strength and ductility of the composite filaments decreased except for in the 6% CPO filament. Due to its acceptable surface morphology and strength, the PLA/CPO filament with 6% CPO was selected for printability testing. The 3D-printed sample of a bone scaffold exhibited good printing quality, demonstrating the potential of the PLA/CPO filament as an improved biocompatible filament for FDM 3D printing.

## 1. Introduction

Today, 3D printing, also known as additive manufacturing (AM), is a revolutionary technology that utilises computer-aided design to print three-dimensional objects layer by layer based on a digital model [1]. This technology has gained widespread use across various industries owing to its numerous advantages over conventional techniques, such as rapid production, the elimination of tooling requirements, high geometrical freedom, and the efficient use of materials [2,3,4]. According to the ISO/ASTM 52,900 standard, 3D printing technologies are classified into seven distinct groups, which include VAT polymerisation, powder bed fusion, binder jetting, material extrusion, direct energy deposition, material jetting, and sheet lamination [5]. Material extrusion is a vital category of additive manufacturing that is commonly employed by researchers, hobbyists, and numerous industries, including pharmaceuticals and biomedical engineering. The technology, originally developed by Stratasys in 1989, is known as Fused Deposition Modelling (FDM) and is used for 3D printing thermoplastic materials [6,7]. Fused Deposition Modelling (FDM) is a popular 3D printing technology because of its versatility and ability to produce functional parts and prototypes from a wide variety of materials [8,9]. It can be utilised to fabricate objects using thermoplastics, metals, ceramics, and composite materials, which has contributed to its widespread adoption [10,11]. In FDM, filaments are heated, softened, and extruded through a hot nozzle and deposited on a building bed layer by layer according to a computer-aided design (CAD) model.

Since filament materials are the backbone of any FDM process, industry and academia have expressed much interest in the production and development of new materials, particularly biocompatible and biodegradable materials that are high quality [11,12,13]. Currently, polylactic acid (PLA), polycaprolactone (PCL), and acrylonitrile butadiene styrene (ABS) have been the most often utilised polymers for filament production [14]. Compared to ABS and PCL, there is a growing demand and promising future for PLA filaments due to their excellent mechanical properties, processability, biodegradability, and biocompatibility, all of which are highly influenced by their stereochemistry and molecular weight [5,15]. Furthermore, PLA’s unique properties make it a viable material with a wide range of industrial uses, including for use in biomedical devices [16,17]. In addition, the US Food and Drug Administration (FDA) has approved it for a number of biomedical and clinical applications [18]. In particular, PLA has been extensively used in bone scaffolds. This is a result of its excellent bioresorption properties, which facilitate its integration with host tissues. PLA has also been combined with other materials to produce FDM filaments for biomedical applications, such as hydroxyapatite (HA), polyethene glycol (PEG), and ferromagnetic materials (Fe_3_O_4_), due to their excellent healing properties [19,20,21,22].

Despite the promising results of using PLA and their composites, an insufficient oxygen supply is a barrier to widely adopting their application in tissue engineering. Several approaches have been used to promote oxygen delivery to bone implants. Growth factors have been added to the implant in order to promote bone neovascularisation. However, it can only be effective with implant sizes of a few millimetres. Other approaches have been adopted to include oxygen-generating materials with the implanted device.

Hydrogen peroxide (H_2_O_2_) has been found to be highly effective in tissue engineering at low concentrations due to its low toxicity, which can be well controlled for tissue engineering applications [23,24]. H_2_O_2_ is generated as an intermediate product during the oxygenation process of oxygen-generating particles, such as calcium peroxide [25,26]. Studies have demonstrated that having an oxygen supply within the scaffold holds great promise for the success of the scaffold’s functionality, as it encourages vascularisation [27]. One study by Hilde et al. [28] fabricated a PLA/CaO_2_ (i.e., calcium peroxide (CPO)) composite bone scaffold via wet solution mixing. They conducted an XTT assay to assess the scaffold cytotoxicity after adding a catalase to the culture medium. The study showed that the incorporation of CaO_2_ particles into biodegradable composite materials made with PLA or PLGA polymers has been found to increase the release of oxygen and to reduce cytotoxicity [28].

Studies found in the biomedical literature have shown that incorporating oxygenation particles into biomedical materials can have a positive impact on bone tissue by promoting vascularisation and regeneration, and by improving the overall healing process. In addition, they also demonstrated that PLA/CPO has potential for use in bone tissue engineering. However, there has been no research on the preparation of PLA/CPO composite filaments for 3D printing using Fused Deposition Modelling (FDM) technology, despite its potential as a promising biomaterial. To address this gap, we developed a novel FDM filament, which was composed of PLA and (CPO, which was produced using a wet solution mixing and hot-melt extruder approach. CPO powder was selected as an oxygen generator due to its proven effectiveness in biomedical applications while remaining non-harmful to the human body. The content of CPO varied in large increments from 3% to 24%, allowing us to examine filament printability and determine the maximum CPO load that can be added. The prepared filaments were characterised in terms of their rheological properties, X-ray diffraction, surface morphology, extrudability, microstructural analysis, mechanical properties, and printability.

## 2. Materials and Methods

### 2.1. Materials

The materials used include: 1.75 MM polylactic acid (PLA) natural filament, which was purchased from Shenzhen eSUN Industrial Co. Ltd., Shenzhen, China; calcium peroxide (CPO) CaO_2_ (−200 mesh size, 75% purity), which was purchased from Sigma-Aldrich (St. Louis, MO, USA); dichloromethane (DCM), which was purchased from Sigma-Aldrich (St. Louis, MO, USA); and deionised water.

### 2.2. Preparation of Composite Filament

To prepare the PLA/CPO composite filaments, 20 g of PLA filaments were cut into small pieces and dissolved in 100 mL of DCM for approximately 30 min at room temperature using a magnetic stirrer set to a speed of 700 rpm. Once the PLA was completely dissolved, CPO powder was added to the solution at different ratios (Table 1) under vigorous stirring for 90 min before it was poured into a large plate to dry for 24 h. After the prepared composite materials were dried, they were cut into small pieces to be loaded into a hot-melt extruder. Figure 1 illustrates the schematic diagram of this process. The extrusion of the composite materials was carried out using a single-screw extruder (King Abdulaziz University, Jeddah, Saudi Arabia) with a nozzle diameter of approximately 2 mm. Our objective was to achieve an optimal filament diameter of 1.75 mm and a smooth surface suitable for commercial Fused Deposition Modelling (FDM) 3D printers. To accomplish this, the composite materials were extruded at various temperatures and feed rates. Experiments were conducted at four different extrusion temperatures, specifically 130 °C, 140 °C, 150 °C, and 155 °C, while maintaining a constant screw speed of 1.5 rpm. The composite was introduced into the extruder at two different feed rates, which are referred to as F1, the feed rate of approximately 1.5 g, and F2, the feed rate of approximately 6.5 g. The extrusion speed was dependent on the feed rates, while the screw speed remained constant at all times.

### 2.3. Filament Characterisation

The JSM-7600F field-emission scanning electron microscope (JEOL, Tokyo, Japan) was employed to examine the surface morphology of samples with different ratios and extrusion conditions. Additionally, optical images of the 3D-printed scaffolds using the composite filament were captured using a Canon 1000D digital camera. Prior to SEM analysis, all samples were sputter-coated with a thin layer of gold using a JFC-1600 auto fine coater (JEOL, Tokyo, Japan). The samples’ elemental compositions were assessed using energy-dispersive X-ray spectroscopy (EDX), which was linked to the SEM. Additionally, an Ultima IV X-ray diffractometer (XRD) (Rigaku, Japan), ICDD (PDF-2/release 2011 RDB), attached with Cu Ka radiations and DB card No. 01-071-4107 were used to observe the material microstructure and phase changes of the samples before and after extruding at a goniometer speed of 1.00 sec and a step of 0.100°.

The samples’ filaments were preheated in a 40 mm cylindrical mould at approximately 230 °C, were compressed manually, and then were allowed to cool to prepare test specimens for rheology testing. The rheological analysis was conducted at room temperature using a discovery HR-3 hybrid rheometer (TA Instruments, New Castle, DE, USA) in a parallel plate configuration (diameter = 40 mm) with a constant gap of 0.5 mm. A flow ramp was performed at 200 °C with a shear rate ranging from 0.1 to 1000 s^−1^.

The mechanical properties of the samples were determined using a universal Instron 3367 testing machine (Norwood, MA, USA) equipped with a 30 kN load cell according to ASTM D4603. Tensile testing was conducted on filament samples ranging in diameter from 1.75 mm to 1.95 mm and with a length of 90 mm using manual grips. The machine was set at a constant crosshead speed of 5 mm/min. The engineering stress–strain curves were used to calculate Young’s modulus (E), yield or ultimate tensile strength (σy), strain at the maximum stress (εm), and strain at break (εb). To ensure accuracy, each experiment was performed three times, and the average was calculated. The information is presented as the mean ± standard deviation (SD). Origin software (OriginPro 8.0, Origin Lab Inc., Northampton, MA, USA) was used to analyse and display the data in the form of graphs.

## 3. Results and Discussion

### 3.1. Optimisation of Extrusion Parameters

To obtain high-quality filaments with a consistent diameter and smooth surface morphology, the extrusion of the PLA/CPO raw material was systematically investigated. The extrusion temperature and feed rate are crucial parameters for achieving optimal results, as shown in Figure 1. The diameter and speed of the extruded filaments were evaluated under different nozzle temperatures (130 °C, 140 °C, 150 °C, and 155 °C) and two sets of feed rates (F1 ≈ 1.5 g and F2 ≈ 6.5 g), as shown in Figure 2. The PLA feedstock was chopped into small pieces to fit the extruder feeder and was loaded simultaneously. The extruder screw speed was maintained at a constant 1.5 rpm throughout the extrusion process. Figure 2 summarises the results for filament diameter and speed at different extrusion conditions.

Figure 2 shows that as the extrusion temperatures decrease, the filament diameters increase regardless of the feed rate. The largest diameter occurred when using a nozzle temperature of 130 °C. The average diameter was 2.10 mm and 2.15 mm when using the feed rates of F1 and F2, respectively. On the other hand, the smallest diameter was found when extruding the filament at a temperature of 155 °C. Filaments with diameters of 1.58 mm and 1.65 mm were obtained when using the feed rates of F1 and F2, respectively. The speed of extrusion is often increased by increasing either the temperature or the feed rates; for example, by increasing the temperature, the viscosity decreases, causing a faster extrusion flow and resulting in a smaller diameter filament. Similarly, with a higher feeding rate, the flow speed increases. However, the results show that the feed rates only affect the filament diameter by ±0.15 mm. This means that the diameter is greatly affected by the extrusion temperature. An optimal diameter of 1.75 mm was achieved using 140-F2, as illustrated in Figure 2 where the extrudability window is highlighted in green with an error of ±0.005, by determining the right temperature range for the desired diameter and then regulating the feed rate for precise results. The findings discussed in the given analysis are consistent with those from previous research on the extrusion of PLA filaments. For instance, a study by Suhaili et al. [29] investigated the effect of extrusion temperature and feed rate on the diameter of 3D-printed filaments. The study found that the extrusion temperature had a significant effect on the filament diameter, while the effect of feed rate was relatively small. Moreover, the study found that decreasing the extrusion temperature resulted in an increase in the filament diameter.

Figure 3 shows SEM images of the produced filaments at different magnifications. Figure 3a shows that at a low temperature of 130 °C, the filament had a large diameter of around 2.15 mm and demonstrated peeling on its surface, as depicted by the red circles. On the other hand, at a high temperature of 155 °C, the filament had an unsymmetrical surface morphology and nonconstant diameters (Figure 3c). In comparison to other filaments, the ideal filament diameter of 1.75 mm achieved at 140 °C had a very smooth surface morphology, and an extruding speed of 1.97 cm/sec allowed for greater control over the extruding process (Figure 3b). At high temperatures, PLA becomes less viscous and the extruding speed is accelerated, resulting in an unsymmetrical filament morphology.

When keeping the temperature and feed rate constant, the extruding speed of the filament decreased as the CPO ratio increased (Table 2). Moreover, a high CPO content resulted in filament accumulation and nozzle clogging, which is a common issue reported in the 3D printing of ABS and graphene composites. For example, in a study involving a ratio of 7.4 wt% of graphene, the 3D printer nozzle became clogged due to the increased graphene content [30]. To ensure a consistent filament diameter, all subsequent samples were extruded at a constant temperature of 140 °C with a feed rate of F2, which had been optimised to achieve the desired diameter of 1.75 mm.

### 3.2. Rheological Properties

The rheological properties of the PLA/CPO composites were analysed and are presented in (Figure 4) in order to assess the viscosity and its suitability for extrusion and 3D printing. As a general rule, materials with a lower viscosity are more suitable for flowing and extrusion, which can improve the quality of 3D printing. Therefore, minimising the viscosity of the material is often desirable to achieve better printing performance. The viscosity-shear rate of the PLA/CPO composite filament samples was measured at 200 °C. All samples exhibited shear-thinning behaviour, which is a typical characteristic of linear polymers and is known as pseudo-plastic fluid behaviour. Moreover, the PLA samples with varying CPO ratios displayed similar shear rates with minor variations in viscosity, most notably for the high and low CPO ratios.

The results demonstrate that all PLA/CPO composite samples exhibit shear-thinning behaviour, which is a typical pseudo-plastic fluid behaviour observed in linear polymers. Interestingly, the viscosity of samples containing a low CPO content (3% and 6%) is quite similar to that of pure PLA [31]. However, samples with higher CPO concentrations (12% and 24%) showed a slightly higher viscosity compared to those with lower concentrations. This suggests that the concentration of CPO in the composite is directly proportional to its viscosity. The increase in viscosity is dependent on various factors, including the concentration, size, distribution, and shape of the filler particles [32]. This suggests that CPO particles disrupt the normal polymer flow, hindering chain segment mobility and making it difficult for the minor phase to disperse evenly in the melt. As a result, higher CPO concentrations can result in poorer dispersion and increased viscosity of the filled polymer. This can be especially problematic during the 3D printing of complex geometries, as it can restrict the deformation of the composite.

### 3.3. Microstructure

Figure 5 illustrates the X-ray diffraction (XRD) patterns of the PLA/CPO composites before and after the extrusion process. These patterns were analysed to confirm the microstructure of the composite material, which can significantly influence its properties. As shown in the figure, the diffraction peak centred at around 16° that corresponds to the PLA indicates its crystalline structure. Additionally, four crystalline peaks were found at 2 θ° of 30°, 35°, 47°, and 53°, corresponding to the CPO particles present in the composites. It was observed that the intensity of CPO peaks increased significantly with the increase in the amount of CPO in the polymeric matrix. The broadening of the XRD peaks of CPO (3%, 6%, 12%, and 24%) was mostly due to the presence of particles in the composites. This broadening became more evident in the XRD pattern after extrusion.

Interestingly, the PLA peak disappeared after extrusion, indicating a transformation from a crystalline to amorphous structure. The XRD patterns for all samples, regardless of the CPO ratios, showed a completely amorphous PLA peak after extrusion. This transformation in the material structure can have significant implications for the mechanical properties of the material. Crystalline polymers have stronger intermolecular bonds, leading to increased strength [33]. However, an amorphous structure can improve bioavailability by increasing the solubility of CPO [33,34]. The transition from a crystalline to amorphous structure observed in the XRD patterns can be attributed to the heating temperature during the extrusion process of the filament [35]. The PLA/CPO composite was melted and extruded, leading to the formation of a new material with a different structure. This transformation can have implications for the mechanical properties of the material, and the next section studies the mechanical properties of the extruded filaments in more detail.

### 3.4. Mechanical Properties

The tensile properties of the PLA/CPO composites were evaluated to investigate the impact of CPO on the mechanical behaviour of the PLA matrix. Moreover, it was examined whether the transformation to an amorphous structure significantly deteriorated the mechanical strength of the composites. Figure 6 and Table 3 show an example of the stress–strain curves and the mechanical properties of PLA/CPO composites with varying CPO contents. Samples prepared using 12% and 24% CPO had linear curves at a low strain followed by plastic deformation in the region of about a 2% strain, while samples prepared using 3% CPO yielded a breaking strain of around 2.8%, which is similar to that for pure PLA. In addition, the tensile strength of the composites varied significantly with increasing CPO concentration.

Samples prepared using 3% and 6% CPO had the highest ultimate strength, whereas samples prepared using 12% and 24% had the lowest ultimate strength. On the other hand, the strain at break decreased as the CPO concentration increased for all CPO concentrations, especially for those with high concentrations. The figure also shows a significant reduction in ultimate strength by approximately 65% for samples with a CPO concentration greater than 6%. The significant decrease in tensile strength and strain seen for samples with a CPO ratio greater than 6% can be attributed to the agglomeration of CPO particles in the polymer matrix, which act as stress concentrators and weaken the composite structure. These results are consistent with those from other studies that have investigated the effect of filler content on the mechanical properties of polymer composites [36]. The study also reported that the experimental error of the strength and Young’s modulus values for 6% CPO is within the range of variations for pure PLA, implying that its strength and Young’s modulus remain unchanged. This finding suggests that 6% CPO can be more suitable for biomedical applications, such as bone scaffolds, where high strength is required [37,38].

### 3.5. Surface Morphology

To ensure the quality of the produced filaments, the surface morphology was carefully examined to confirm that they were consistent and smooth. The findings of this investigation demonstrate that increasing the CPO ratio results in a rough and irregular surface morphology of the filaments, which is clearly demonstrated in Figure 7. Further analysis of SEM images and surface texture indicates that filaments with higher CPO ratios of 24% and 12% exhibit rougher surfaces compared to those with lower CPO ratios of 6% and 3%. Moreover, the study highlights that an increase in CPO concentration, especially above 6%, leads to a reduction in filament ductility, as depicted in Figure 6. This reduction in ductility causes the filaments to become brittle, rendering them unsuitable for FDM 3D printing. The inflexible filaments cannot be fed through the feeder/tubing for extrusion and CPO particles can block the nozzle head, as observed in previous studies [39]. Additionally, the reduced strength in samples with a CPO concentration higher than 6% resulted in fragile filaments that may crack while being fed through the nozzle head, leading to further issues in the printing process [40].

Previous studies have suggested that reducing the distance between the filament feed inlet and the extruder nozzle can be helpful in overcoming the issue of feeding the filament through the pipes/tubing [36]. However, this solution does not address the low strength issue associated with using filaments with high CPO ratios. Based on the results obtained in the previous sections, it can be concluded that using a PLA filament with a CPO concentration of 6% is a favourable choice. This is because, at this concentration, the composite exhibited not only a smooth surface morphology but also better viscosity and improved mechanical properties such as strength and ductility.

To demonstrate the printability of filaments with a CPO concentration of 6%, a proof-of-concept scaffold object was fabricated using a commercial Fused Deposition Modelling (FDM) 3D printer; specifically, the Creality Ender 3 Pro (manufactured by Shenzhen Creality 3D Technology Co., LTD., Shenzhen, China) was used. The scaffold was designed to have a square shape with dimensions of 8 × 8 mm and a height of 1.5 mm, and with a porosity of 25% and a pore size of 0.60 mm. The printer bed was maintained at 60 °C, and the nozzle temperature was set at 200 °C.

Figure 8 presents an optical image of the 3D-printed scaffold using a CPO concentration of 6%. The image demonstrates the successful printing of the scaffold and the good quality of the optimised CPO content. The optimised CPO concentration allowed for the production of a scaffold with smooth and uniform surfaces, indicating good printability. The geometry shows a well-structured scaffold with a strong and robust structure. Overall, the results suggest that PLA filaments with a 6% CPO concentration could be utilised to produce high-quality and functional bone scaffolds for biomedical applications. This concentration ensures good printing quality, strength, and roughness while avoiding issues such as blockages, nozzle head clogs, and filament breakage, making it an ideal choice for FDM 3D printing applications.

## 4. Conclusions

The study aimed to develop PLA/CPO composite filaments through wet solution mixing and extrusion for biomedical applications. The extrusion process was optimised to achieve filament quality and a diameter of 1.75 mm. Viscosity analysis showed similar results for all samples, with slightly higher viscosities for those with a higher CPO content. Samples with a CPO content over 6% showed a significant reduction in ultimate strength, while those with 6% CPO had the highest ultimate strength of 56.9 MPa. SEM analysis of the samples revealed that filaments with a lower CPO content had a smoother surface and were more ductile, making them suitable for FDM. All samples exhibited a change in microstructure from crystalline to amorphous after extrusion while still retaining CPO viability. The composite material filaments with a 6% CPO concentration had comparable mechanical strength to pure PLA and consistent surface roughness. These filaments were used to print samples using a commercial FDM 3D printer and showed good printability. These findings suggest that PLA/CPO composite material filaments with a CPO content of 6% can be used to fabricate bone scaffolds with suitable mechanical properties and an acceptable surface morphology for biomedical applications. Further research is required to determine the maximum CPO content that can be loaded into the composite filament without compromising its flexibility. Additionally, the research highlights the importance of conducting biological studies to evaluate the potential biocompatibility and safety of the composite filament for medical applications.

## Figures and Tables

**Figure 1 polymers-15-02229-f001:**
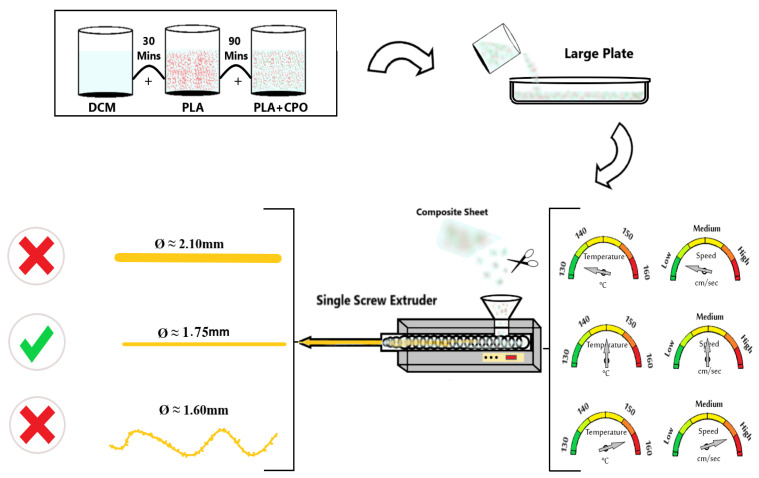
Schematic illustration of composite filament preparation.

**Figure 2 polymers-15-02229-f002:**
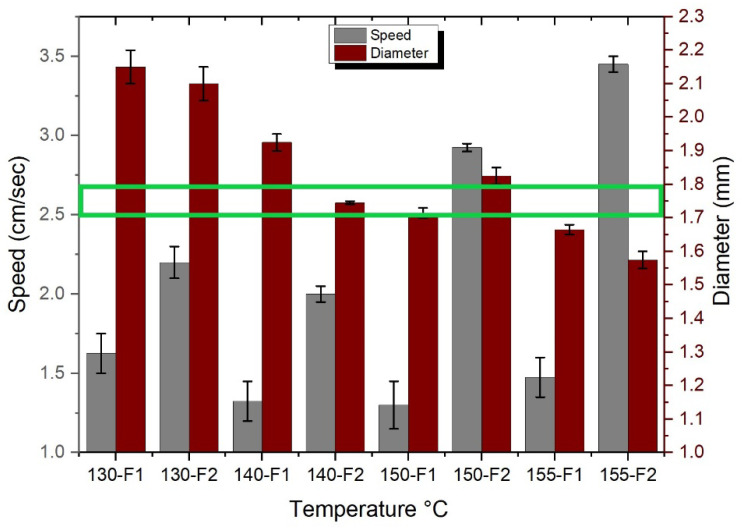
Extrudability window of PLA material (i.e., green square).

**Figure 3 polymers-15-02229-f003:**
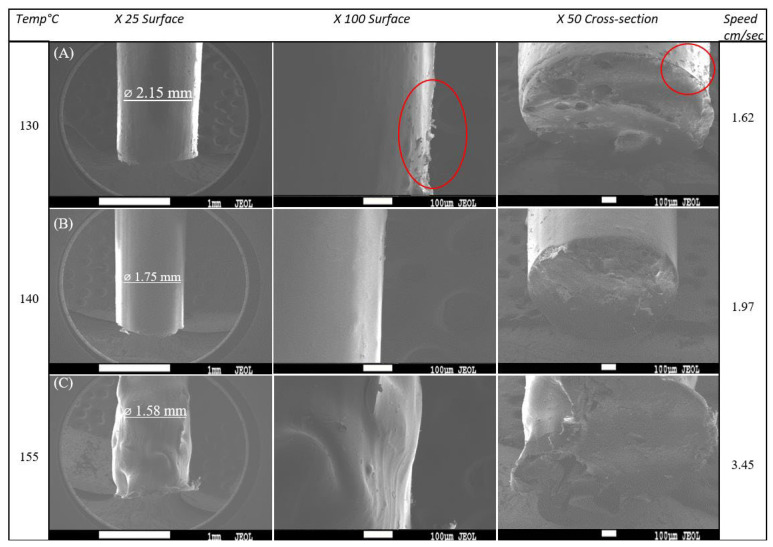
SEM images of extruded filaments at (**A**) 130 °C, (**B**) 140 °C, and (**C**) 155 °C. Surface peeling is depicted by red circles.

**Figure 4 polymers-15-02229-f004:**
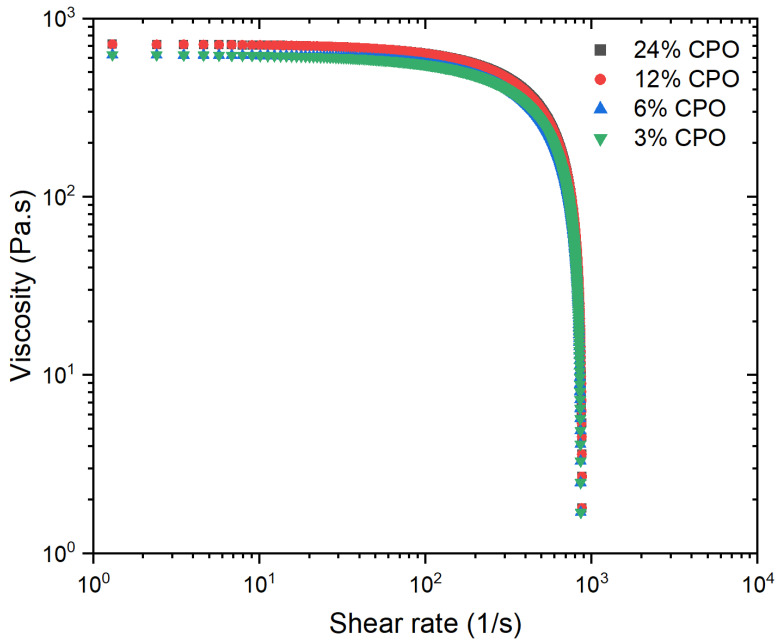
Viscosity-shear rate for the PLA/CPO composite filament samples.

**Figure 5 polymers-15-02229-f005:**
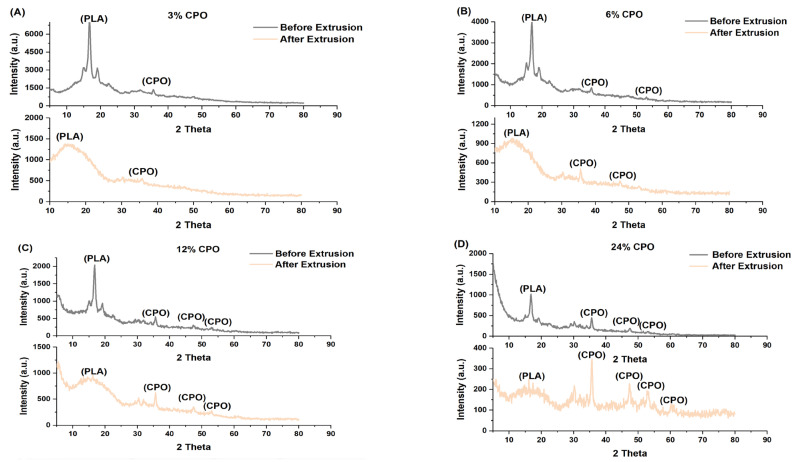
X-ray diffraction (XRD) patterns of PLA/CPO composites before and after hot-melt extrusion: (**A**) 3% CPO, (**B**) 6% CPO, (**C**) 12% CPO, and (**D**) 24% CPO.

**Figure 6 polymers-15-02229-f006:**
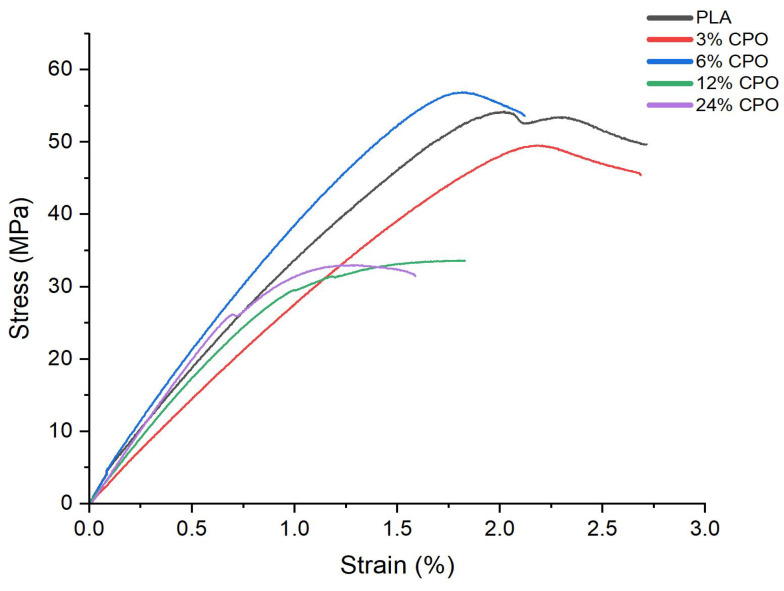
Stress–strain curves of PLA/CPO composites.

**Figure 7 polymers-15-02229-f007:**
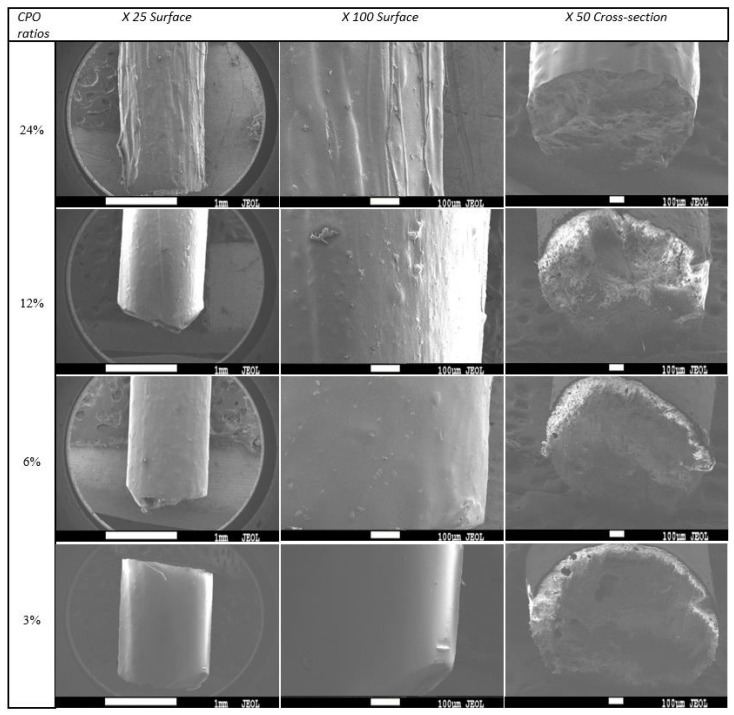
SEM images of PLA and CPO composite filaments at different ratios.

**Figure 8 polymers-15-02229-f008:**
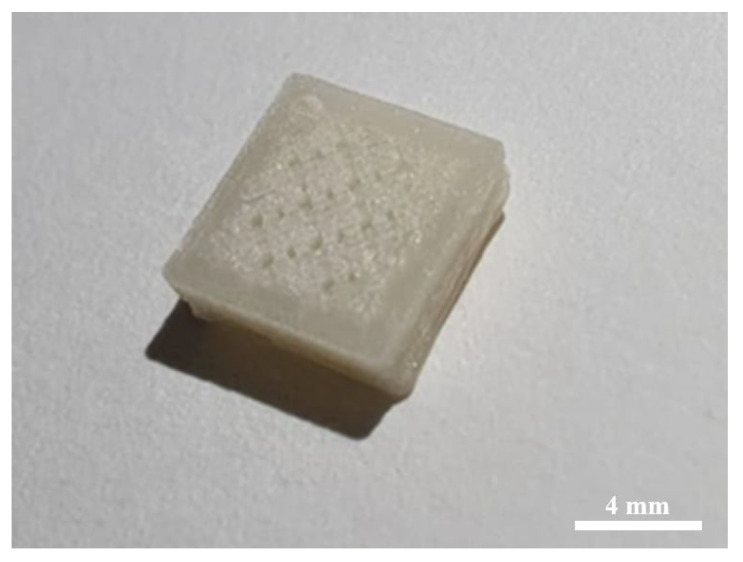
Optical image of the 3D-printed scaffold using CPO of 6% at a magnification of ×10.

**Table 1 polymers-15-02229-t001:** PLA/CPO composite ratios.

Sample No.	Sample Name	PLA (%wt.)	CPO (%wt.)
1	3% CPO	97	3
2	6% CPO	94	6
3	12% CPO	88	12
4	24% CPO	76	24

**Table 2 polymers-15-02229-t002:** Speed of extrusion for different CPO ratios.

CPO Ratio (%)	Speed (cm/Sec)
3% CPO	9.5
6% CPO	6
12% CPO	4.25
24% CPO	2.75

**Table 3 polymers-15-02229-t003:** Mechanical properties of PLA/CPO composites with standard deviation.

CPO Ratios	Tensile Strength σ_m_, MPa	Strain at Break ε_b_, %	Young’s Modulus E, GPa
0%	52.2 ± 2.1	2.7 ± 0.25	3.5 ± 0.32
24%	32.9 ± 1.5	1.6 ± 0.28	4.1 ± 0.45
12%	33.6 ± 1.4	2.0 ± 0.26	3.3 ± 0.38
6%	55.8 ± 1.8	2.0 ± 0.22	4.1 ± 0.36
3%	49.5 ± 2.0	2.7 ± 0.25	2.9 ± 0.31

## Data Availability

Data is contained within the article. For additional information please contact the corresponding author.

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
