# Peer review of "Preparation of Polylactic Acid/Calcium Peroxide Composite Filaments for Fused Deposition Modelling"

_polymers, 2023, doi:10.3390/polym15092229_

Round 1

Reviewer 1 Report (Previous Reviewer 2)

Dear,

In this new version of the manuscript, the authors revised and corrected the suggestions, improving the quality. Questions were answered satisfactorily. In addition, the recommendations were accommodated in the manuscript. In view of this, the manuscript became clearer, more technical, and consistent with the objectives. Therefore, I recommend the publication that has been reviewed. Here are some small suggestions before publication:

> Page 6. XRD. Please inform the type of radiation (CuKα?), the speed of the goniometer, and the step (0.02°?);

> Page 6. Characterisations. Please add the optical microscope methodology (equipment, model, bright or dark field);

> Page 6. Characterisations = Characterizations

> Figure 8. Add the scale on the MO micrograph, as well as the magnification used;

Yours sincerely,

Author Response

Reviewer #1:

In this new version of the manuscript, the authors revised and corrected the suggestions, improving the quality. Questions were answered satisfactorily. In addition, the recommendations were accommodated in the manuscript. In view of this, the manuscript became clearer, more technical, and consistent with the objectives. Therefore, I recommend the publication that has been reviewed. Here are some small suggestions before publication:

Comment 1:

Page 6. XRD. Please inform the type of radiation (CuKα?), the speed of the goniometer, and the step (0.02°?);

Response 1:

We have added the required XRD information in our revised manuscript in line (135-137)

Additionally, an Ultima IV X-ray diffractometer (XRD) (Rigaku, Japan), ICDD (PDF-2/release 2011 RDB), attached with Cu Ka radiations and DB card No. 01-071-4107 were used to observe the material microstructure and phase changes of the samples before and after extruding at goniometer speed of 1.00 sec and step of 0.100°.

Comment 2:

Page 6. Characterisations. Please add the optical microscope methodology (equipment, model, bright or dark field);

Response 2:

We have added the used camera for the optical images in the methodology section.

Comment 3:

Page 6. Characterisations = Characterizations                                                                                                                                

Response 3:

We have corrected the subtitle as required.

Comment 4:

Figure 8. Add the scale on the MO micrograph, as well as the magnification used;

Response 4:

We have added the scale bar and magnification used.

Reviewer 2 Report (Previous Reviewer 1)

Th authors have answered the questions well and have made significant revisions to the manuscript. The manuscript should be publishable in current form.

Author Response

Reviewer #2:

Th authors have answered the questions well and have made significant revisions to the manuscript. The manuscript should be publishable in current form.

Response:

Thank you.

Reviewer 3 Report (New Reviewer)

The manuscript under review examines the creation of a new biomaterial filament by using wet solution mixing and extrusion to prepare a Poly Lactic Acid (PLA)/Calcium Peroxide (CPO) composite. The authors analyze the characteristics of the filaments through X-ray diffraction analysis, surface morphology assessment, filament extrudability evaluation, microstructural analysis, and examination of their rheological and mechanical properties. The study found that the mechanical strength and ductility of the composite filaments decreased. While the study is valuable and deserves publication, there are some significant concerns that need to be addressed to improve the manuscript's readability and clarity before accepting it for publication.

1-    The use of English language is reasonable, however, there are a number of punctuation and grammatical errors; that should be corrected and rephrased using academic English for a better flow of text for reader. There are also some statements need an enhancement; for example, in the abstract the authors said,” In this study, we aimed to develop a new biomaterial filament by preparing Poly Lactic Acid (PLA)/Calcium Peroxide (CPO) composite using wet solution mixing and extrusion.”. It is unusual to use (we or I) inside the scientific manuscript.

2-    According to the authors, the PLA/CPO samples showed microstructural changes from crystalline to amorphous, which indicates a problem with the composite preparation. Typically, in composites, there is no chemical reaction that occurs, and each material maintains its own microstructure. As a result, the composite should exhibit the peaks of both materials. However, in this case, a chemical reaction occurred, causing the PLA's microstructure to change from crystalline to amorphous. Therefore, it may be inappropriate to use the term "composite" in this paper.

3-    Based on comment two, it is recommended to analyze the prepared filament using FTIR for more understanding of what occurred during the preparation process.

4-    The authors used a CPO fractions (3-24%). It was not clear for it is a weight ratio or volume ratio till I reached table. It is recommended to make it obvious from the beginning in the abstract.

5-    It is unclear why the authors chose the current weight fractions for the study. The selection of weight fractions 3, 6, 12, and 24 is not explained in the manuscript, and these are relatively large steps. In composites, authors typically select a very low loading fraction up to 1%, a low loading fraction between 1 to 10%, or a high loading fraction of more than 10%. However, in this case, it is not clear why the authors chose these specific weight fractions, and the reasoning behind their selection is not provided.

6-    The random selection of the additive weight fraction was reflected on the mechanical results. It is obvious there is a large deviation in the stress strain figure between the samples performance.

7-    The introduction is poor and needs a lot of enhancement. The introduction references are very old. Although we are now in 2023, there are no references in 2023, and only one reference in 2022. I think incorporating updated references show the interesting of researchers on the paper topic. Here are some recommended papers:

https://doi.org/10.3390/polym14235299

https://doi.org/10.1016/j.compstruct.2023.116992

https://doi.org/10.1016/j.compstruct.2022.116386

10.1016/j.compositesa.2023.107513

https://doi.org/10.1016/j.matlet.2022.133543

https://doi.org/10.3390/polym14163321

https://www.nature.com/articles/s41598-023-30300-z

8-    In the characterization section, Testing standards need to be provided. 

9-    There is an error in the figure numbering. Stress-strain curve had a number 1, although it is figure 6.

10-                  The dispersion of the Calcium Peroxide within the PLA after mixing. Did the authors study it? SEM-images of neat PLA and PLA with CPO after mixing need to add this article.

11-    Results are merely described and is limited to comparing the experimental observation and describing results. The authors are encouraged to include a more detailed results and discussion section and critically discuss the observations from this investigation with existing literature.

12-   Figure 8 can’t illustrate anything regarding the quality of the product. I think there is no need for this figure and its details inside the manuscript.

13-   Conclusions need to be compacted to highlight the outcomings of the scientific paper. Furthermore, the authors didn't mention their future work at the end of the conclusions.

 Please, read the text carefully before the next submission of the paper.

Author Response

Reviewer #3:

The manuscript under review examines the creation of a new biomaterial filament by using wet solution mixing and extrusion to prepare a Poly Lactic Acid (PLA)/Calcium Peroxide (CPO) composite. The authors analyze the characteristics of the filaments through X-ray diffraction analysis, surface morphology assessment, filament extrudability evaluation, microstructural analysis, and examination of their rheological and mechanical properties. The study found that the mechanical strength and ductility of the composite filaments decreased. While the study is valuable and deserves publication, there are some significant concerns that need to be addressed to improve the manuscript's readability and clarity before accepting it for publication.

Comment 1:

The use of English language is reasonable, however, there are a number of punctuation and grammatical errors; that should be corrected and rephrased using academic English for a better flow of text for reader. There are also some statements need an enhancement; for example, in the abstract the authors said,” In this study, we aimed to develop a new biomaterial filament by preparing Poly Lactic Acid (PLA)/Calcium Peroxide (CPO) composite using wet solution mixing and extrusion.”. It is unusual to use (we or I) inside the scientific manuscript

Response 1:

We have revised the manuscript for punctuation and grammar and have avoided using 'we' or 'I'. for example: the above “we” statement in the abstract was modified to the following:

In this study, a new biomaterial filament was developed by preparing Poly Lactic Acid (PLA)/Calcium Peroxide (CPO) composite using wet solution mixing and extrusion.

Comment 2:

According to the authors, the PLA/CPO samples showed microstructural changes from crystalline to amorphous, which indicates a problem with the composite preparation. Typically, in composites, there is no chemical reaction that occurs, and each material maintains its own microstructure. As a result, the composite should exhibit the peaks of both materials. However, in this case, a chemical reaction occurred, causing the PLA's microstructure to change from crystalline to amorphous. Therefore, it may be inappropriate to use the term "composite" in this paper.

Response 2:

The XRD results section revealed that all samples exhibited peaks of both materials prior to extrusion, indicating that no reaction occurred during the mixing process, as demonstrated in Figure 5. However, a change in the microstructure was observed specifically in the PLA matrix after hot-melt extrusion, as depicted in Figure 5. This transformation in the microstructure can be attributed to the heating temperature during filament extrusion, as mentioned in lines (255-257) and supported by reference (27). Additionally, the presence of CPO peaks after extrusion confirms that no reaction occurred between the two materials and only the PLA matrix was affected by the extrusion temperature.

The transition from crystalline to amorphous structure observed in the XRD patterns can be attributed to the heating temperature during the extrusion process of the filament [27].

Comment 3:

Based on comment two, it is recommended to analyze the prepared filament using FTIR for more understanding of what occurred during the preparation process.

Response 3:

As previously stated, we can confidently confirm that no changes occurred in the microstructure of the composite due to reactions, and each constituent remained unaltered. However, alterations were observed in the polymer matrix following the hot-melt extrusion process. Nevertheless, these modifications did not adversely affect the required mechanical properties.

Comment 4:

The authors used a CPO fractions (3-24%). It was not clear for it is a weight ratio or volume ratio till I reached table. It is recommended to make it obvious from the beginning in the abstract.

Response 4:

We have added this to the abstract in our revised manuscript.

Comment 5:

It is unclear why the authors chose the current weight fractions for the study. The selection of weight fractions 3, 6, 12, and 24 is not explained in the manuscript, and these are relatively large steps. In composites, authors typically select a very low loading fraction up to 1%, a low loading fraction between 1 to 10%, or a high loading fraction of more than 10%. However, in this case, it is not clear why the authors chose these specific weight fractions, and the reasoning behind their selection is not provided.

Response 5:

The selected weight fractions were chosen based on the need to identify the maximum CPO load that could be added to the filament while ensuring printability. We agree that the weight fractions selected in our study are relatively large steps, but we needed to ensure that the composite filaments were printable without any issues. We have now added this explanation to the manuscript in lines (92-93) to provide a better understanding of our rationale for the selection of weight fractions. We hope that this clarifies any confusion and addresses your concerns.

The content of CPO was varied in large increments from 3% to 24% to examine filament printability and determine the maximum CPO load that can be added. The prepared filaments were characterized in terms of their rheological properties, x-ray diffraction, surface morphology, extrudability, microstructural analysis, mechanical properties, and printability.

Comment 6:

The random selection of the additive weight fraction was reflected on the mechanical results. It is obvious there is a large deviation in the stress strain figure between the samples performance.

Response 6:

We agree with your observation that there is a significant deviation in the stress-strain curves of the samples due to the random selection of the additive weight fraction. However, we also noticed a distinct change in the stress-strain curve between ratios below 6% and those above 12%. This finding suggests that there may be an opportunity for future work to investigate different ratios below 12% and identify the highest possible ratio of CPO/PLA with good mechanical properties. We have added this discussion to the manuscript to highlight this point and emphasize the importance of investigating different ratios below 12%. We hope that this addresses your concerns and provides more insight into the potential for future research.

Further research is needed to determine the maximum CPO contents that can be loaded into the composite filament without compromising its flexibility, which is essential for practical use. Additionally, it is important to conduct biological studies to explore the potential biocompatibility and safety of the composite filament for medical applications.

Comment 7:

The introduction is poor and needs a lot of enhancement. The introduction references are very old. Although we are now in 2023, there are no references in 2023, and only one reference in 2022. I think incorporating updated references show the interesting of researchers on the paper topic. Here are some recommended papers:

https://doi.org/10.3390/polym14235299

https://doi.org/10.1016/j.compstruct.2023.116992

https://doi.org/10.1016/j.compstruct.2022.116386

https://doi.org/10.1016/j.compositesa.2023.107513

https://doi.org/10.1016/j.matlet.2022.133543

https://doi.org/10.3390/polym14163321

https://www.nature.com/articles/s41598-023-30300-z

Response 7:

We have included the recommended papers in our revised manuscript.

[1]        A. Fouly, I. A. Alnaser, A. K. Assaifan, and H. S. Abdo, "Evaluating the Performance of 3D-Printed PLA Reinforced with Date Pit Particles for Its Suitability as an Acetabular Liner in Artificial Hip Joints," Polymers, vol. 14, no. 16, p. 3321, 2022. [Online]. Available: https://www.mdpi.com/2073-4360/14/16/3321.

[7]        M. Zarei et al., "Enhanced bone tissue regeneration using a 3D-printed poly(lactic acid)/Ti6Al4V composite scaffold with plasma treatment modification," Scientific Reports, vol. 13, no. 1, p. 3139, 2023/02/23 2023, doi: 10.1038/s41598-023-30300-z.

[9]        J. Crespo-Miguel, D. Garcia-Gonzalez, G. Robles, M. Hossain, J. M. Martinez-Tarifa, and A. Arias, "Thermo-electro-mechanical aging and degradation of conductive 3D printed PLA/CB composite," Composite Structures, p. 116992, 2023/04/03/ 2023, doi: https://doi.org/10.1016/j.compstruct.2023.116992.

[13]      Y. Chen, T. Lu, L. Li, H. Zhang, H. Wang, and F. Ke, "Fully biodegradable PLA composite with improved mechanical properties via 3D printing," Materials Letters, vol. 331, p. 133543, 2023/01/15/ 2023, doi: https://doi.org/10.1016/j.matlet.2022.133543.

[17]      A. Fouly, A. K. Assaifan, I. A. Alnaser, O. A. Hussein, and H. S. Abdo, "Evaluating the Mechanical and Tribological Properties of 3D Printed Polylactic-Acid (PLA) Green-Composite for Artificial Implant: Hip Joint Case Study," Polymers, vol. 14, no. 23, p. 5299, 2022. [Online]. Available: https://www.mdpi.com/2073-4360/14/23/5299.

[22]      A. Akmal Zia et al., "Mechanical and energy absorption behaviors of 3D printed continuous carbon/Kevlar hybrid thread reinforced PLA composites," Composite Structures, vol. 303, p. 116386, 2023/01/01/ 2023, doi: https://doi.org/10.1016/j.compstruct.2022.116386.

Comment 8:

In the characterization section, Testing standards need to be provided.

Response 8:

We have added the mechanical testing standard in line (144-145).

Comment 9:

There is an error in the figure numbering. Stress-strain curve had a number 1, although it is figure 6.

Response 9:

We have fixed the error.

Comment 10:

The dispersion of the Calcium Peroxide within the PLA after mixing. Did the authors study it? SEM-images of neat PLA and PLA with CPO after mixing need to add this article.

Response 10:

Thank you for your comment. As we are developing a new composite filament for 3D printing, there are certainly many aspects that could be studied. In this particular study, we focused on the mixing, extrudability, rheological properties, XRD, mechanical properties, surface properties, and proof of concept of 3D printing. However, we understand that the dispersion of Calcium Peroxide within PLA after mixing is an important aspect that was not covered in this study. We did conduct another study where we investigated the dispersion, porosity, antibacterial activity, and oxygen release of the material, but that research was included in a separate paper that has been accepted by one of the pharmaceutics journals. We will continue to explore and improve our understanding of this material, and we appreciate your interest and feedback.

Comment 11:

Results are merely described and is limited to comparing the experimental observation and describing results. The authors are encouraged to include a more detailed results and discussion section and critically discuss the observations from this investigation with existing literature.

Response 11:

Thank you for your comment. As we mentioned before and as we are developing a new composite filament, there was there is no existing literature on this specific study. However, where applicable we improved the discussion by including references from relevant topics. Below are the literature that we included in the results and discussion section:

[27 ]C.-C. Kuo, J.-Y. Chen, and Y.-H. Chang, "Optimization of Process Parameters for Fabricating Polylactic Acid Filaments Using Design of Experiments Approach," Polymers, vol. 13, p. 1222, 04/09 2021.

[28]         U. Kalsoom, P. N. Nesterenko, and B. Paull, "Recent developments in 3D printable composite materials," RSC Advances, 10.1039/C6RA11334F vol. 6, no. 65, pp. 60355-60371, 2016.

[29]         A. Duhduh, H. Noor, A. Kundu, and J. Coulter, Advanced Additive Manufacturing of Functionally Gradient Multi Material Polymer Components with Single Extrusion Head: Melt Rheology Analysis. 2019.

[30]         D. Shumigin, E. Tarasova, A. Krumme, and P. Meier, "Rheological and mechanical properties of poly (lactic) acid/cellulose and LDPE/cellulose composites," Materials Science, vol. 17, no. 1, pp. 32-37, 2011.

[31]         D. S. Frank and A. J. Matzger, "Effect of Polymer Hydrophobicity on the Stability of Amorphous Solid Dispersions and Supersaturated Solutions of a Hydrophobic Pharmaceutical," Molecular Pharmaceutics, vol. 16, no. 2, pp. 682-688, 2019/02/04 2019.

[32]         X. Wang, H. C. Schröder, and W. E. G. Müller, "Amorphous polyphosphate, a smart bioinspired nano-/bio-material for bone and cartilage regeneration: towards a new paradigm in tissue engineering," Journal of Materials Chemistry B, 10.1039/C8TB00241J vol. 6, no. 16, pp. 2385-2412, 2018.

[33]         Y. Kobayashi, T. Ueda, A. Ishigami, and H. Ito, "Changes in Crystal Structure and Accelerated Hydrolytic Degradation of Polylactic Acid in High Humidity," (in eng), Polymers (Basel), vol. 13, no. 24, Dec 10 2021.

[34]         C. Dong, I. J. Davies, C. C. M. Fornari Junior, and R. Scaffaro, "Mechanical properties of Macadamia nutshell powder and PLA bio-composites," Australian Journal of Mechanical Engineering, vol. 15, no. 3, pp. 150-156, 2017/09/02 2017.

[35]         G. Eddy, E. Poinern, R. Brundavanam, and D. Fawcett, "Nanometre Scale Hydroxyapatite Ceramics for Bone Tissue Engineering," American Journal of Biomedical Engineering, vol. 2013, pp. 148-168, 08/27 2013.

[36]         J. Pakkanen, D. Manfredi, P. Minetola, and L. Iuliano, About the Use of Recycled or Biodegradable Filaments for Sustainability of 3D Printing. 2017, pp. 776-785.

[37]         T. Beran, T. Mulholland, F. Henning, N. Rudolph, and T. A. Osswald, "Nozzle clogging factors during fused filament fabrication of spherical particle filled polymers," Additive Manufacturing, vol. 23, pp. 206-214, 2018/10/01/ 2018.

[38]         M. Bahraminasab, "Challenges on optimization of 3D-printed bone scaffolds," BioMedical Engineering OnLine, vol. 19, no. 1, p. 69, 2020/09/03 2020.

Comment 12:

Figure 8 can’t illustrate anything regarding the quality of the product. I think there is no need for this figure and its details inside the manuscript.

Response 12:

The purpose of including this figure is to provide a validation of the printability of the filament using an FDM machine as suggested by other reviewers. While it may not directly assess the quality of the product, we believe that it serves as a proof of concept and demonstrates the feasibility of using this filament for 3D printing applications.

Comment 13:

Conclusions need to be compacted to highlight the outcomings of the scientific paper. Furthermore, the authors didn't mention their future work at the end of the conclusions.

Response 13:

We have compacted the conclusion to highlight the outcome and added the future work in line (354-355). Words counts is 231 including the addition of future work.

Round 2

Reviewer 3 Report (New Reviewer)

Many thanks for the revision and for incorporating all suggested changes to the manuscript that are nicely reflected. The authors did a good job of improving the article. I believe that the article has become much better, and now I recommend this article for publication.

This manuscript is a resubmission of an earlier submission. The following is a list of the peer review reports and author responses from that submission.

Round 1

Reviewer 1 Report

This work focuses on the fabrication of PLA/CPO composites for 3D printing of bone scaffolds. The authors varied the composition of PLA and CPO, with four different formulations. Subsequently, the surface/cross-sectional morphology, rheological properties, microstructure and mechanical properties were characterized. There are several major inadequacies in this paper that should be addressed.

11)     Two major themes of manuscript other than PLA and CPO is: a) 3D-printing (FDM) and b) Tissue engineering (Bone scaffold). These themes took up a significant portion of the abstract, introduction and keywords sections. However, there was not demonstration of 3D-printing and the bone scaffold in the manuscript. In that case, the hypothesis remains unproven.

22)     The mechanical properties (strength) were stated to be enhanced by 6% of CPO, and the authors claim that it makes them excellent materials for 3D-printing of bone scaffolds. Firstly, the increase in strength is very minimal, from 54.2 MPa to 56.9 MPa. Secondly, this could just be a statistical error, as the authors only submitted 1 set of mechanical properties for each composition. There are not standard deviation (or error bars) to the properties. Thirdly, how does increase in strength help in the application as a bone scaffold? Why not ductility?

33)     The rheological properties (mainly shear-thinning behavior) is a known behavior of thermoplastics at elevated temperature. This purpose of the characterization was also not discussed. Was it for 3D-printing?

44)     The purpose of many characterizations was not explicitly spelt out. For example, the XRD. How the crystalline of amorphous PLA affects the application or 3D-printing?

55)     There were many formatting errors. The XRD figures appear twice in pg 8. There were missing in-text reference to figures. There was a “PLA” labelled under the column of CPO ratios in table 3. Figure 3 SEM images table was poorly formatted.

Author Response

All comments have been answered thoroughly.

Reviewer 2 Report

Dear,

The authors developed PLA/calcium peroxide composites for application as a biomaterial. However, the manuscript needs to address the focus of the title, as well as indicate whether the results are suitable for biomaterial. In addition, it would be interesting to report the cytotoxicity of the composites prepared. Some details should be reviewed:

> Introduction. Authors need to speed up the introduction on the topic, specifically: (a) add a specific review; (b) show the cytotoxicity behavior of calcium peroxide;

> Review the entire manuscript as it appears: Error! Reference source not found

> The composites developed are for medical application. Why did the authors not perform specific assays for biomaterials?

> Results and discussion need comparison with literature data;

> Are the results obtained suitable for application as a biomaterial? Manuscript data needs to be approached from an application point of view;

Author Response

we have answered all queries.
